# Impact of Thyroid Function on Pregnancy and Neonatal Outcome in Women with and without PCOS

**DOI:** 10.3390/biomedicines10040750

**Published:** 2022-03-23

**Authors:** Sarah Feigl, Barbara Obermayer-Pietsch, Philipp Klaritsch, Gudrun Pregartner, Sereina Annik Herzog, Elisabeth Lerchbaum, Christian Trummer, Stefan Pilz, Martina Kollmann

**Affiliations:** 1Division of Obstetrics and Maternal Fetal Medicine, Department of Obstetrics and Gynecology, Medical University of Graz, 8036 Graz, Austria; s.feigl@medunigraz.at (S.F.); philipp.klaritsch@medunigraz.at (P.K.); 2Division of Endocrinology and Diabetology, Department of Internal Medicine, Medical University of Graz, 8036 Graz, Austria; barbara.obermayer@medunigraz.at (B.O.-P.); elisabeth.lerchbaum@medunigraz.at (E.L.); christian.trummer@medunigraz.at (C.T.); stefan.pilz@medunigraz.at (S.P.); 3Institute for Medical Informatics, Statistics and Documentation (IMI), Medical University of Graz, 8036 Graz, Austria; gudrun.pregartner@medunigraz.at (G.P.); sereina.herzog@medunigraz.at (S.A.H.)

**Keywords:** polycystic ovary syndrome (PCOS), thyroid disorders in pregnancy, autoimmune thyroid disease

## Abstract

Background: Women with polycystic ovary syndrome (PCOS) are more prone to autoimmune thyroiditis, and both disorders lead to subfertility and pregnancy-related complications. The aim of this study was to investigate whether mothers with and without PCOS and their offspring have comparable thyroid parameters at term and how thyroid parameters are associated with perinatal outcome in this population. Methods: This cross-sectional observational study was performed in a single academic tertiary hospital in Austria. Seventy-nine pregnant women with PCOS and 354 pregnant women without PCOS were included. Blood samples were taken from the mother and cord blood at birth. Primary outcome parameters were maternal and neonatal thyroid parameters at delivery. Secondary outcome parameters were the composite complication rate per woman and per neonate. Results: Thyroid dysfunction was more prevalent among PCOS women (*p* < 0.001). At time of birth, free triiodothyronine (fT3) levels were significantly lower in PCOS than in non-PCOS women (*p* = 0.005). PCOS women and their neonates had significantly higher thyreoperoxidase antibody (TPO-AB) levels (*p* = 0.001). Women with elevated TPO-AB had a significantly higher prevalence of hypothyroidism (*p* < 0.001). There was a significant positive correlation between maternal and neonatal free thyroxine, fT3 and TPO-AB levels. There were no significant differences in thyroid parameters between women or neonates with or without complications. Conclusions: Our results demonstrate a higher prevalence of thyroid dysfunction and autoimmunity in PCOS women, supporting a common etiology of both disorders. We were not able to show an association between complication rate and thyroid parameters.

## 1. Introduction

Polycystic ovary syndrome (PCOS) and thyroid malfunction influence fertility and impact the health of mother and child during pregnancy. It has been demonstrated that both conditions can lead to pregnancy-related disorders [1,2,3,4,5,6,7]. We have previously shown that the risk for maternal complications during pregnancy is increased in PCOS patients as compared to healthy controls [3]. Reported complications include gestational diabetes, pregnancy-induced hypertension (PIH), pre-eclampsia, preterm delivery, and increased need for caesarean sections [1,3,4,8].

Both PCOS and thyroid malfunction including autoimmune thyroid disease (AITD) are common disorders in women of reproductive age, given that 11% of women are affected by AITD and around 1–3% suffer from hypo- or hyperthyroidism [6].

Thyroid malfunction and AITD are more frequent in PCOS patients, at least in non-pregnant cohorts [9,10]. In pregnant women, the prevalence of overt and subclinical hypothyroidism ranges from 0.3–0.5% to 4–17%, respectively, and thyreoperoxidase antibodies (TPO-AB) or thyroglobulin antibodies (Tg-AB) are found in 2% to 18% [7,11,12,13].

PCOS women are at higher risk for infertility and pregnancy complications, even more so due to their disposition for thyroid disease. Elevated TPO-AB as well as subclinical and overt hypothyroidism can lead to infertility and adverse pregnancy outcomes, such as miscarriage and preterm birth, Small for Gestational Age (SGA), pre-eclampsia, stillbirth, cesarean delivery, and impaired neuro-intellectual development of the fetus [5,6,11,14,15,16,17,18,19,20,21,22]. Still, data on pregnancy outcome in women with both PCOS and thyroid malfunction is scarce. Concerning the common ground, a connection between PCOS, hypothyroidism, and autoimmune disorder has already been established [23,24,25,26,27,28,29,30,31]. The etiology of both disorders might include an immune modulating effect of estrogen and a genetic predisposition, since polymorphisms in PCO-related gene coding for fibrillin 3 may also predispose women to AITD by altering immune tolerance [30]. Due to the lack of data on PCOS women, their thyroid function, and pregnancy outcomes, we looked at thyroid parameters of PCOS women in their third trimester and cord blood thyroid parameters of their neonates and compared them to a non-PCOS control group. Neonatal thyroid screening in Austria is usually done by capillary blood sampling in the first days of life. Due to its better availability and to keep the time difference between maternal and neonatal blood collection as small as possible, we performed cord blood sampling. Thyroid parameters in cord blood seem to be as reliable as neonatal capillary testing [32,33]. 

The aim of this study was to investigate whether mothers with and without PCOS and their offspring have comparable thyroid parameters at term and how thyroid parameters are associated with perinatal outcome in this population.

## 2. Materials and Methods

### 2.1. Study Design/Setting

This cross-sectional study was performed between March 2013 and December 2015 at a single academic tertiary hospital (Department of Obstetrics and Gynecology, Medical University of Graz, Austria).

Seventy-nine pregnant women with PCOS, according to the ESHRE/ASRM 2003 criteria [34], and 354 pregnant women without PCOS with singletons ≥37 + 0 weeks of gestation were included. All participants were 18 years or older at time of inclusion. Written informed consent was provided by all participants. Exclusion criteria were severe comorbidities (neurodegenerative, immune mediated, cardiovascular, or infectious disease), suspected abnormal placentation (placenta accreta, increta, or percreta), placenta previa, previous vertical uterine incision, a history of major abdominal surgery, or known fetal malformations.

Ethical approval was obtained by the ethical review board (ethics committee at the Medical University of Graz, Austria; 24-179ex11/12).

### 2.2. Outcome Measures

Primary outcome parameters for this report were maternal and neonatal thyroid parameters (thyroid stimulating hormone (TSH), free triiodothyronine (fT3), free thyroxine (fT4), and TPO-AB) at delivery. In 23 PCOS women, thyroid parameters were additionally evaluated before conception and at regular intervals during pregnancy and after birth. Secondary outcome parameters were the composite complication rate per woman (a binary composite outcome consisting of gestational diabetes, PIH, pre-eclampsia, and operative delivery) and per neonate (a binary composite outcome consisting of SGA, large for gestational age (LGA), fetal growth restriction (FGR), fetal acidosis, and intensive care unit admission).

### 2.3. Data Sources/Measurement

Blood samples were taken from the mother and umbilical cord within 5 min after delivery. Additionally, blood samples of 23 PCOS women were obtained at seven time points (preconception, pregnancy weeks 12–14, 20–22, 24–28, 34, at birth and 6 months postpartum). For the neonatal sample, mixed umbilical cord blood was used. Laboratory kits and assays did not change from 2013 to 2015. TSH was measured using ADVIA Centaur and XP immunoassays (Siemens, Erlangen, Germany). Levels between 0.1 and 4.0 µU/mL were considered normal. FT4 was measured with ADVIA Centaur fT4, while 9.5–24 pmol/L was defined as the normal range. For fT3, ADVIA Centaur fT3 was used, with 3–6.3 pmol/L as the normal range. TPO-AB were detected using Immulite 2000 Anti TPO (Siemens, Erlangen, Germany). Up to 60 U/mL are regarded as normal. Definitions for gestational diabetes, PIH, pre-eclampsia, operative delivery, SGA, LGA, FGR, and fetal acidosis have been previously published [35]. Subclinical hyperthyroidism was defined as TSH below 0.1 µU/mL and normal fT3 and fT4. Overt hyperthyroidism was defined as TSH below 0.1 µU/mL and fT3 above 6.3 pmol/L and/or fT4 above 24 pmol/L. Subclinical hypothyroidism was defined as TSH above 4.0 µU/mL and fT3 and fT4 within the normal range. Overt hypothyroidism was defined as TSH above 4.0 µU/mL and fT3 below 3 pmol/L and/or fT4 below 9.5 pmol/L [36]. Demographic data were extracted from the local perinatal database (PIA, ViewPoint, GE Healthcare, Solingen, Germany) and the medical documentation system or patient files. Information on pre-pregnancy thyroid function was collected at admission using a questionnaire. 

### 2.4. Sample Size Calculation

It was estimated that 350 non-PCOS and 35 PCOS patients were sufficient to detect differences between the two groups for effect sizes of at least 0.5, with a significance level of 5% and a power of 80%. As we expected some dropouts and difficulties with cord blood analysis, we aimed to include at least 400 patients. A data quality check after one year of recruitment revealed that more patients than expected had to be excluded due to comorbidities and that cord blood analysis was not feasible in some cases due to insufficient material. 

### 2.5. Statistical Methods

The current analysis is a cross-sectional analysis of a previously published prospective cohort study [35]. Metric parameters were summarized using median and range (minimum to maximum) and categorical parameters as absolute and relative frequencies. Differences between groups were analyzed using Mann–Whitney U or Fisher’s exact test. Correlation between maternal and neonatal parameters was calculated using Spearman correlation coefficient. All statistical analyses were conducted using R Software Version 3.5.3 and 3.6.1. *p* values < 0.05 were considered statistically significant. The results of any subgroup analysis should be interpreted in an exploratory fashion.

## 3. Results

### 3.1. Participants

Overall, 499 women were assessed for eligibility and 433 were included for analysis (79 (18%) PCOS and 354 (82%) non-PCOS women). A flow chart of participants was published before as well as demographic data, which showed no statistically significant differences between the groups [35]. However, significantly more PCOS women had preexistent thyroid dysfunction and levothyroxine (LT) therapy before conception compared to non-PCOS women. All cases of known preexisting dysfunctions were hypothyroidism (Table 1).

### 3.2. Thyroid Dysfunction at Time of Birth 

Thyroid dysfunction at time of birth was comparable between the groups with and without PCOS (Table 2). Only one case of subclinical hyperthyroid function was seen (non-PCOS). Eight PCOS and 58 non-PCOS women had subclinical hypothyroidism (*p* = 0.410). Overt hypothyroidism was diagnosed in two and six PCOS and non-PCOS women, respectively (*p* = 0.633). Of all women with hypothyroid function at birth, 21 had preexistent LT therapy (3 PCOS, 18 non-PCOS women).

### 3.3. Thyroid Parameters

TSH levels did not show significant differences between PCOS/non-PCOS women and their respective neonates (Table 2). TSH levels in neonates of all mothers with or without hypothyroidism were comparable (PCOS median (range): 8.4 (3.8–53.7) and non-PCOS: 7.7 (1.7–47.5); *p* = 0.221). In subgroup analyses, neonates of PCOS women with hypothyroidism had significantly higher TSH levels compared to neonates of PCOS women without hypothyroidism (PCOS median (range): 10.3 (5.3–53.7) and non-PCOS: 6.8 (1.7–37.2); *p* = 0.016). Neonates of non-PCOS mothers with and without hypothyroid function did not show a statistically significant difference in TSH levels (PCOS median (range): 7.9 (3.8–21.9) and non-PCOS: 7.8 (2.0–47.5); *p* = 0.879). Analysis of longitudinal data showed stable TSH throughout pregnancy.

FT4 levels were not significantly different in women with or without PCOS or their neonates (Table 2). 

In PCOS women, fT3 levels were significantly lower than in non-PCOS women (*p* = 0.005).

In neonates, fT3 levels were not significantly different in PCOS and non-PCOS groups (*p* = 0.055). FT3 and fT4 did not differ between neonates of mothers with or without hypothyroidism, neither in pooled nor in subgroup analyses of male and female neonates. 

Longitudinal data of both fT3 and fT4 levels showed a decrease throughout pregnancy.

PCOS women had significantly higher TPO antibody levels as compared to non-PCOS women (Table 2). Also, comparing all neonates, those of PCOS women had higher TPO antibody levels compared to neonates of non-PCOS women (*p* = 0.049). In subgroup analyses comparing male and female neonates separately, no significant difference was found for either sex. 

Among the 13 women with TPO-AB >60 U/mL, prevalence of hypothyroidism was significantly higher than in women with TPO antibodies in the normal range (61.5% versus 13.8%; *p* < 0.001). However, TSH levels, fT3 and fT4 in neonates of women with TPO-AB >60 U/mL and in neonates of women with normal TPO-AB were not different (*p* = 0.833, *p* = 0.255, and *p* = 0.610, respectively). Comparing neonates with TPO-AB >60 (*n* = 11) and <60 U/mL (*n* = 346), no differences in other thyroid parameters were found (*p* = 0.374 for TSH, *p* = 0.526 for fT3, and *p* = 0.777 for fT4).

Longitudinal data showed a decrease of TPO-AB throughout pregnancy and slight increase after delivery (Figure 1).

### 3.4. Correlation of Maternal and Neonatal Parameters

In the total cohort, there was a significant positive correlation between maternal and neonatal fT4 levels (r = 0.43, *p* < 0.001), fT3 levels (r = 0.17, *p* = 0.002), and TPO-AB levels (r = 0.76, *p* < 0.001). No significant correlation was found between maternal and neonatal TSH levels (r = 0.05, *p* = 0.328). In subgroup analyses of the PCOS cohort, there was also a significant correlation seen in fT4 levels (r = 0.40, *p* = 0.003) and TPO-AB levels (r = 0.67, *p* < 0.001), but not for ft3 levels (r = 0.22, *p* = 0.099). In the non-PCOS cohort, ft4 (r = 0.43, *p* < 0.001), ft3 (r = 0.14, *p* = 0.017), and TPO-AB (r = 0.79, *p* < 0.001) were significantly positively correlated (Figure 2).

### 3.5. Association of Thyroid Parameters at Birth and Perinatal Outcome

There were no significant differences in thyroid parameters between women or neonates with and without complications (Table 3). The complication rate between women with hypothyroidism and women with euthyroid function and their respective neonates was not significantly different (Table 4).

## 4. Discussion

Our results demonstrate a higher prevalence of thyroid dysfunction and autoimmunity in PCOS women, supporting a common etiology of both disorders.

In our cohort, elevated TPO-AB levels were more prevalent in PCOS women and their neonates compared to controls. Elevation of neonatal TPO-AB was not associated with change in other thyroid parameters, but women with elevated TPO-AB showed a significantly higher prevalence of hypothyroidism.

Higher prevalence of autoimmunity in PCOS patients has been reported before, and as described above, lower immune tolerance and genetic disposition play an important role in its development [30]. Still, the causal effect of elevated TPO-AB leading to infertility and pregnancy complications is unclear. One hypothesis is a direct harmful effect of TPO-AB on ovarian tissue or hypothyroidism caused by autoimmunity, leading to infertility [6,20,37]. 

Ft3 levels were significantly lower in PCOS women compared to non-PCOS controls, and neonates of PCOS women with hypothyroidism had significantly higher TSH values as compared to neonates of PCOS women without hypothyroidism, indicating diminished hormonal supply during pregnancy. 

We found a positive correlation of TPO-AB between mothers and neonates in both groups, due to passing of the placental barrier. In our analysis, TPO-AB levels in cord blood were not associated with neonatal complications. In other studies, elevated TPO-AB in cord blood at delivery were also not associated with neonatal thyroid dysfunction [32,38,39,40].

No correlation was seen in maternal and neonatal TSH, as can also be expected since TSH does not pass the placental barrier. We demonstrated a positive correlation between maternal and neonatal fT4 and fT3. At birth, fT4 and fT3 in cord blood mainly originate from the fetus [38]. These hormones do not cross the placental barrier as easily as TPO-AB, but maternal thyroid hormones have been detected in fetal tissue and a positive correlation between maternal fT4 and fT3 in the third trimester and in cord blood has been proposed before and is in line with our finding [41].

Although at time of birth, there was no difference in thyroid dysfunction between PCOS and non-PCOS women, the high number of hypothyroidism in all women at delivery implies a lack of thyroid monitoring during pregnancy. More PCOS women had pre-diagnosed hypothyroidism and especially in cases with pre-existent LT therapy, therefore monitoring is important. Recommendations for thyroid screening and LT therapy before and during pregnancy are varying. Screening all women seeking pregnancy produces a lot of health care costs, hence in some countries only women at higher risk (e.g., infertility, symptoms of thyroid disease, age > 30) are being screened [7,40,42]. Without question, women undergoing ART or those with preexistent LT therapy should have frequent TSH monitoring before and every 4–6 weeks during pregnancy, due to their higher demand of thyroid hormones in the first weeks of pregnancy [40,43,44,45]. Screening for thyroid autoimmunity should be performed in PCOS women, due to their elevated risk for developing thyroid dysfunction, pregnancy disorders, and postpartum thyroiditis. Beneficial effects of LT supplementation on pregnancy loss have been proven for women with hypothyroidism and autoimmunity [27,46,47]. 

Our longitudinal data of 23 PCOS women show a decrease of TPO-AB and fT3 and fT4 during pregnancy, while TSH remains stable. This can be explained by physiological changes during pregnancy like higher demand of fT3 and fT4 until relevant amounts of hormones are produced by the fetal thyroid gland in the early second trimester, and higher immune tolerance during pregnancy. It is well known that thyroid disorders may remain unrecognized until pregnancy-modulated changes or investigation of causes for infertility bring them to light [36]. Although most of these changes seen are due to physiological adaptions, like the decrease of TSH in early pregnancy due to interactions with human chorionic gonadotropin (HCG), disorders of the thyroid gland, such as autoimmune malfunction, might also be triggered [2].

Perinatal complications were more prevalent in the PCOS cohort compared to the non-PCOS cohort. There were no significant differences in thyroid parameters between women or neonates with and without complications. Regarding our outcome parameters, TPO-AB, TSH, and fT3 and fT4 did not seem to have an effect on maternal or neonatal complication rate. Comparing women with and without hypothyroidism, we did not detect a difference in complication rate, although it has been stated that hypothyroidism is linked with preterm delivery and lower Apgar scores and FGR, as well as pre-eclampsia, which is more frequent in women with thyroid dysfunction [48,49,50]. 

So far, published data on neonatal thyroid levels and the perinatal complication rate of women with thyroid dysfunction are either retrospective analyses or prospective samples of small cohorts [16,17]. Our data contribute to a better understanding of thyroid function in PCOS women during pregnancy. 

## 5. Conclusions

We found a higher prevalence of hypothyroidism and autoimmunity in PCOS women and a high prevalence of hypothyroidism in all women at delivery even with pre-existent LT therapy. We suggest that thyroid evaluation and screening for thyroid autoimmunity should be performed in PCOS women presenting at a fertility clinic as recommended in ASRM guidelines [45]. Monitoring throughout pregnancy is important, especially in cases with pre-existent LT therapy.

## Figures and Tables

**Figure 1 biomedicines-10-00750-f001:**
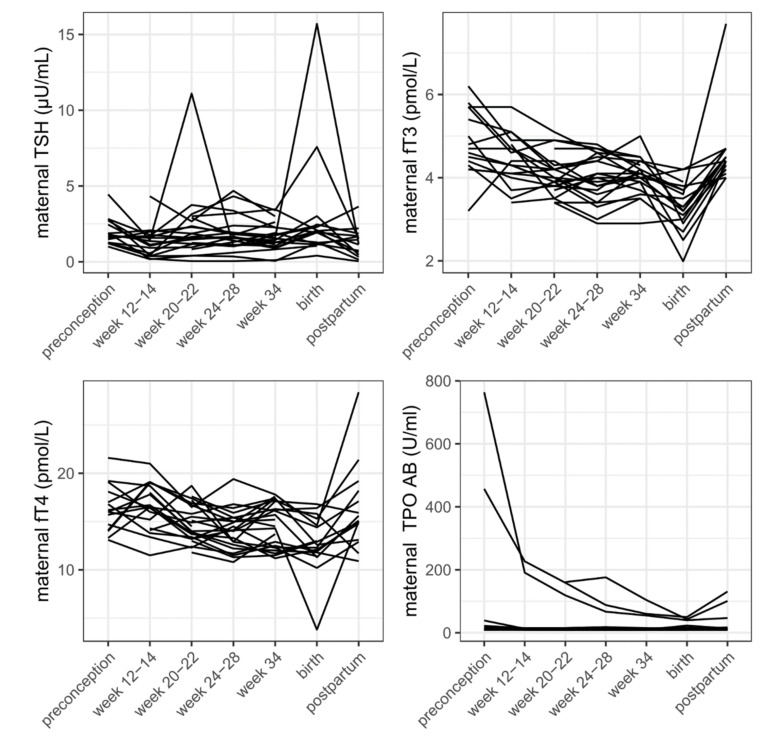
TSH, fT3, fT4, and TPO-AB in PCOS women before, throughout, and after pregnancy.

**Figure 2 biomedicines-10-00750-f002:**
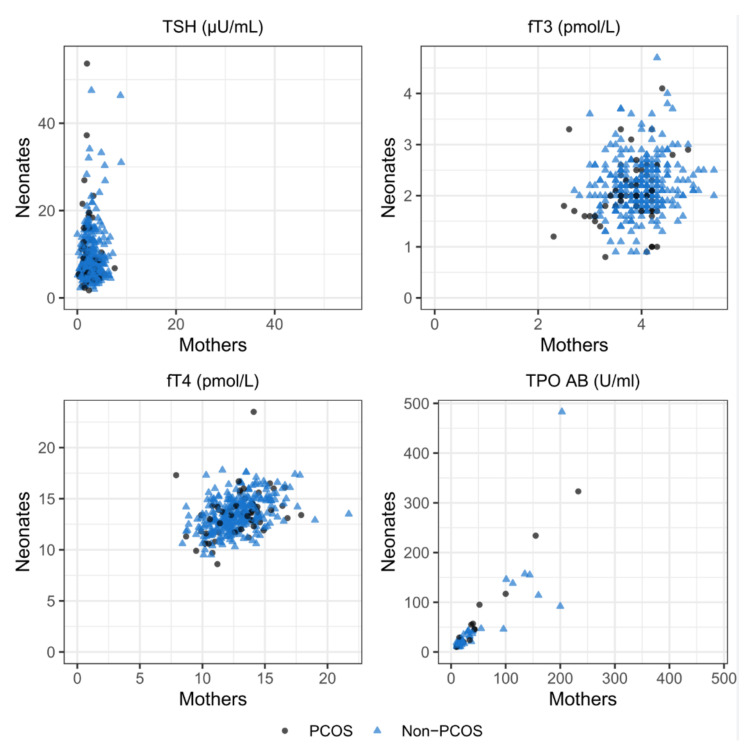
Correlation of maternal and neonatal TSH, fT3, fT4, and TPO-AB levels.

**Table 1 biomedicines-10-00750-t001:** Number of thyroid dysfunction and thyroid medication received before pregnancy in PCOS and non-PCOS women. Values presented are *n* (%).

	PCOS (*n* = 79)	Non-PCOS (*n* = 354)	*p*-Value
Thyroid dysfunction	24 (30.4)	42 (11.9)	<0.001
Thyroid medication	30 (38.0)	57 (16.1)	<0.001

**Table 2 biomedicines-10-00750-t002:** Thyroid parameters in PCOS and non-PCOS women and their neonates. Values shown are median (range). (TSH: µU/mL, fT3 and ft4: pmol/L TPO-AB: U/mL with 9.9 U/mL as minimal detectable value). Between 10.7% (non-PCOS) and 16.5% (PCOS) of maternal values were missing, whereas 151 (non-PCOS) and 27 (PCOS) as well as 149 (non-PCOS) and 29 (PCOS) values were available for female and male neonates, respectively.).

		PCOS Women (*n* = 79)	Non-PCOS Women (*n* = 354)	*p*-Value
TSH		2.3 (0.3–7.6)	2.6 (0.0–8.9)	0.126
FT4		12.9 (7.9–17.9)	12.6 (8.4–21.7)	0.881
FT3		3.8 (2.3–4.9)	4.0 (2.5–5.4)	0.005
TPO-AB		9.9 (9.9–233)	9.9 (9.9–203)	0.001
		PCOS Neonates (*n* = 79)	Non-PCOS Neonates (*n* = 354)	
TSH	Total	8.1 (1.7–53.7)	7.8 (2.0–47.5)	0.769
	Male	8.9 (3.6–53.7)	7.1 (2.0–46.4)	0.421
	Female	8.0 (1.7–26.9)	7.9 (2.3–47.5)	0.720
FT4	Total	13.5 (8.6–23.5)	13.2 (9.5–17.8)	0.899
	Male	13.4 (9.7–23.5)	13.2 (9.5–17.8)	0.665
	Female	13.6 (8.6–17.3)	13.3 (9.5–17.6)	0.808
FT3	Total	2.0 (0.8–4.1)	2.1 (0.9–4.7)	0.055
	Male	2.0 (0.8–4.1)	2.2 (0.9–3.8)	0.058
	Female	2.0 (0.9–3.3)	2.1 (0.9–4.7)	0.418
TPO-AB	Total	9.9 (9.9–323)	9.9 (9.9–483)	0.049
	Male	9.9 (9.9–323)	9.9 (9.9–157)	0.078
	Female	9.9 (9.9–46)	9.9 (9.9–483)	0.312

**Table 3 biomedicines-10-00750-t003:** Comparison of maternal and neonatal thyroid values in mothers and neonates with and without complications. Values shown are median (range). (TSH: µU/mL, fT3 and ft4: pmol/L, TPO-AB: U/mL with 9.9 U/mL as minimal detectable value).

	Perinatal Complications (*n* = 96)	No Perinatal Complications (*n* = 337)	*p*-Value
Maternal TSH	2.5 (0.1–6.7)	2.6 (0.0–8.9)	0.425
Neonatal TSH	9.1 (2.6–46.4)	7.5 (1.7–53.7)	0.051
Maternal fT4	12.5 (7.9–21.7)	12.8 (8.4–19.0)	0.132
Neonatal fT4	13.3 (9.7–17.8)	13.3 (8.6–23.5)	0.638
Maternal fT3	4.0 (2.3–5.4)	3.9 (2.7–5.4)	0.551
Neonatal fT3	2.2 (0.9–4.0)	2.1 (0.8–4.7)	0.600
Maternal TPO-AB	9.9 (9.9–200)	9.9 (9.9–233)	0.077
Neonatal TPO-AB	9.9 (9.9–323)	9.9 (9.9–483)	0.319

**Table 4 biomedicines-10-00750-t004:** Comparison of maternal and neonatal complication rate in hypothyroid and euthyroid women. Values shown are *n* (%).

	Hypothyroidism (*n* = 66)	Euthyroid Function (*n* = 316)	*p*-Value
Maternal complication	35 (53.0)	164 (51.9)	0.893
Neonatal complication	19 (28.8)	66 (20.9)	0.192

## Data Availability

The datasets used and analyzed during the current study are available from the corresponding author upon reasonable request.

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
