# Peer review of "Impact of Thyroid Function on Pregnancy and Neonatal Outcome in Women with and without PCOS"

_biomedicines, 2022, doi:10.3390/biomedicines10040750_

Round 1

Reviewer 1 Report

Introduction: Well done study.

Material and methods: It is a cross-sectional study was performed between March 2013 and December 2015, Seventy-nine pregnant women with PCOS, and 354 pregnant women without PCOS. Primary outcome parameters for this report were maternal and neonatal thyroid parameters (thyroid-stimulating hormone (TSH), free triiodothyronine (fT3), free thyroxine (fT4), and TPO-AB) at delivery. 

Please add references for the sample size calculations. 

Results:  showed that TSH levels did not show significant differences between PCOS/non-PCOS women and their respective neonates. FT4 levels were not significantly different in women with or without PCOS or their neonates. In PCOS women, fT3 levels were significantly lower than in non-PCOS women. In neonates, fT3 levels were not significantly different in PCOS and non-PCOS groups. 

PCOS women had significantly higher TPO antibody levels as compared to nonPCOS women. Please add references to this statement. 

Discussion: Without question, women undergoing ART or those with preexistent LT therapy should have frequent TSH monitoring before and during pregnancy due to their higher demand for thyroid hormones in the first weeks of pregnancy. Please add the time interval and references. 

Conclusion: We suggest that thyroid evaluation and screening for thyroid autoimmunity should be performed in PCOS women presenting at a fertility clinic. As per ASRM, it is recommended for every patient undergoing fertility evaluation. 

Author Response

Thank you very much for considering our manuscript entitled “Impact of thyroid function on pregnancy and neonatal outcome in women with and without PCOS“ suitable for publication in Biomedicines. We took care to respond to the comments and adopted the manuscript according to the suggestions.
  1. Please add references for the sample size calculations. 

Thank you for this important comment. We report on a cross-sectional analysis of a previously published prospective cohort study. The initial sample size estimation states that for any comparison between PCOS and non-PCOS patients the power to detect an effect size (for example Cohen's d, defined as the difference between two means divided by a standard deviation) of at least 0.5 is 80%, irrespective of the parameter investigated. However, the results of any subgroup analyses should be interpreted in an exploratory fashion. To make this clear we added respective sentences in the statistical methods section.

  1. PCOS women had significantly higher TPO antibody levels as compared to nonPCOS women. Please add references to this statement. 

Thank you for this comment. This statement describes our results, and the respective data is shown in Table 2, a cross-reference has been placed at the end of the statement. 

  1. Without question, women undergoing ART or those with preexistent LT therapy should have frequent TSH monitoring before and during pregnancy due to their higher demand for thyroid hormones in the first weeks of pregnancy. Please add the time interval and references.  

Thank you for this valuable comment. We added references to this statement. Women with preexistent LT therapy are at higher risk of developing hypothyroidism during pregnancy and we suggest check-ups before pregnancy and every 4-6 weeks during pregnancy, and this was added to the manuscript.

  1. We suggest that thyroid evaluation and screening for thyroid autoimmunity should be performed in PCOS women presenting at a fertility clinic. As per ASRM, it is recommended for every patient undergoing fertility evaluation. 

Thank you for this comment, we adapted the paragraph and added the ASRM guidelines as reference.

Reviewer 2 Report

Journal: Biomedicines (ISSN 2227-9059)

Manuscript ID: biomedicines-1652633

Type: Article

Title: Impact of thyroid function on pregnancy and neonatal outcome in women with and without PCOS

Authors: Sarah Feigl , Barbara Obermayer-Pietsch , Philipp Klaritsch , Gudrun Pregartner , Sereina Annik Herzog , Elisabeth Lerchbaum , Christian Trummer , Stefan Pilz , Martina Kollmann *

Section: Endocrinology and Metabolism Research

Special Issue: Molecular Research on Polycystic Ovary Syndrome (PCOS)

Revision:

The Article entitled: "Impact of thyroid function on pregnancy and neonatal outcome in women with and without PCOS" investigated whether mothers with and without Polycystic ovary syndrome (PCOS) and their offspring have comparable thyroid parameters and how thyroid parameters are associated with perinatal outcome. The author used a wide range of samples suitable for the study. Statistical analysis is also very thorough and adequate. The article is very interesting and I think that the manuscript is suitable for its publication in Biomedicines journal.

Author Response

Thank you very much for considering our manuscript entitled “Impact of thyroid function on pregnancy and neonatal outcome in women with and without PCOS“ suitable for publication in Biomedicines.